# Reduced Serum Circulation of Cell-Free DNA Following Chemotherapy in Breast Cancer Patients

**DOI:** 10.3390/medsci9020037

**Published:** 2021-05-25

**Authors:** Evelyn Adusei, John Ahenkorah, Nii Armah Adu-Aryee, Kevin Kofi Adutwum-Ofosu, Emmanuel Ayitey Tagoe, Nii Koney-Kwaku Koney, Emmanuel Nkansah, Nii Ayite Aryee, Richard Michael Blay, Bismarck Afedo Hottor, Joe-Nat Clegg-Lamptey, Benjamin Arko-Boham

**Affiliations:** 1Department of Anatomy, University of Ghana Medical School, University of Ghana, Accra P.O. Box GP 4236, Ghana; evelynadusei8@gmail.com (E.A.); jahenkorah@ug.edu.gh (J.A.); KAdutwum-Ofosu@ug.edu.gh (K.K.A.-O.); nkkoney@ug.edu.gh (N.K.-K.K.); eonkansah002@st.ug.edu.gh (E.N.); rmblay@ug.edu.gh (R.M.B.); bakhottor@ug.edu.gh (B.A.H.); 2Department of Surgery, University of Ghana Medical School, University of Ghana, Accra P.O. Box GP 4236, Ghana; naduaryee@chs.edu.gh (N.A.A.-A.); clegglamptey@ug.edu.gh (J.-N.C.-L.); 3Department of Surgery, Korle-Bu Teaching Hospital, Korle Bu, Accra P.O. Box 77, Ghana; 4Department of Medical Laboratory Sciences, School of Biomedical and Allied Health Sciences, University of Ghana, Accra P.O. Box KB 143, Ghana; eatagoe@chs.edu.gh; 5Department of Medical Biochemistry, University of Ghana Medical School, University of Ghana, Accra P.O. Box GP 4236, Ghana; drnaryee@yahoo.co.uk

**Keywords:** circulating cell-free DNA, cfDNA, breast cancer, chemotherapy, biomarker

## Abstract

Breast cancer is the most common malignancy in women, with alarming mortalities. Neoadjuvant treatments employ chemotherapy to shrink tumours to a well-defined size for a better surgical outcome. The current means of assessing effectiveness of chemotherapy management are imprecise. We previously showed that breast cancer patients have higher serum circulating cell-free DNA concentrations. cfDNA is degraded cellular DNA fragments released into the bloodstream. We further report on the utility of cfDNA in assessing the response to chemotherapy and its potential as a monitoring biomarker. A total of 32 newly diagnosed and treatment-naive female breast cancer patients and 32 healthy females as controls were included. Anthropometric, demographic and clinicopathological information of participants were recorded. Each participant donated 5 mL of venous blood from which sera were separated. Blood sampling was carried out before the commencement of chemotherapy (timepoint 1) and after the third cycle of chemotherapy (timepoint 2). qPCR was performed on the sera to quantify ALU 115 and 247 levels, and DNA integrity (ALU247/ALU115) was determined. ALU 115 and 247 levels were elevated in cancer patients but were significantly decreased after the third cycle of chemotherapy (T2) compared to T1. DNA integrity increased after the third cycle. Serum cfDNA may provide a relatively inexpensive and minimally invasive procedure to evaluate the response to chemotherapy in breast cancer.

## 1. Introduction

Cancer incidence and mortality is on the ascendancy and has become a major public health concern. Currently, cancer is the second leading cause of death globally, and was responsible for about 9.6 million deaths in 2018 [1]. In recent years, cancer incidence and mortality have seen a steady rise on the African continent, with breast cancer among the leading causes of cancer deaths in females [2]. The WHO attributes this trend to increased life expectancy, increased urbanization and the adoption of western lifestyles [3]. Globally, breast cancer is the world’s most common cancer in women [4], a trend equally witnessed in Ghana, accompanied by high mortality rate [5,6]. The biggest challenge to the successful management and treatment of breast cancer in Ghana and other low- and middle-income countries (LMICs) is late detection. In Ghana and most African countries, the disease is diagnosed late and associated with high mortality rates and is currently the leading cause of cancer-related deaths among women in Ghana [4]. Despite recent advances in the care and management of the disease, morbidity and mortality rates in Ghana are still unacceptably high [7,8].

As much as public education on the disease has been intensified to create the necessary awareness and early detection, there is still the need to explore other reliable scientific means of early detection, and the accurate and reliable prediction of prognosis to influence the management of the disease, especially in low resource settings such as Ghana. The universally existing therapies of surgery, chemotherapy, radiotherapy and hormone therapy [9] are employed in treatment and/or management of the disease. In countries such as Ghana where many patients report to health facilities with metastatic disease, chemotherapy is the first and principal treatment option [10,11], with the aim of a systemic clearance of the cancerous cells to prolong life. For those with locally advanced disease, chemotherapy is again first introduced with the aim of shrinking the tumour to a well-defined size for a better surgical outcome [12,13]. Chemotherapy is given as a combination of drugs and given in cycles, usually at three-week intervals [14]. There is not one standard chemotherapeutic regimen that works for all patients and therefore, clinicians assess the effectiveness of chemotherapy on a case-to-case basis for a determination. In cases where patients are unresponsive to particular drugs, the drugs are changed, or the dosage reviewed [14,15]. The assessment of the effectiveness of chemotherapy is commonly carried out following the third cycle of treatment (usually about 9 weeks) from the commencement and sometimes may require repeated biopsies for histopathological confirmation of tumour response to treatment [16,17]. This procedure comes with inherent challenges, as delays (unintentional) in the administration of suboptimal drugs may lead to disease progression and worsened prognosis. Another challenge is that the size of the tumour after chemotherapy may not accurately reflect the tumour response and may lead to incorrect clinical decisions [18]. This warrants the urgent need to develop new alternatives and more reliable means of assessing the effectiveness of chemotherapy in breast cancer management.

Interest in circulating cell free DNA (cfDNA) and the prospects of its utility in cancer diagnosis, control and monitoring continues to grow. cfDNA, degraded DNA fragments released into the bloodstream [19], was first discovered in 1948 [20] and was subsequently observed in autoimmune cells [21] and later in cancers [22]. cfDNA presents as ALU (Arthrobacter luteus) sequences which are transposable repeated elements belonging to the short interspersed nuclear elements (SINEs) abundant elements in the human genome, with a copy number of 1.4 × 106 per genome [23]. They are typically about 300 nucleotides in length. ALU elements multiply within the genome in a retroposition process through RNA polymerase III-derived transcripts from evolution [24]. The source of cfDNA in healthy individuals is solely by apoptosis, producing evenly sized shorter DNA fragments (ALU 115), but in cancers, however, necrosis contributes uneven longer DNA fragments (ALU 247) to the shorter fragments from apoptosis [22,23,24,25,26]. Therefore, elevated levels of longer fragments of DNA in the bloodstream has been targeted as a good marker for the presence of malignant tumour DNA [27,28,29,30]. DNA integrity, the ratio of longer to shorter fragments, has been explored for its usefulness in the diagnosis and prognosis of cancers. It has been suggested to be increased in cancer patients and particularly in metastatic cases than in non-metastatic cases. It has been found to predict tumour progression, and regional lymph node metastases in primary breast cancer patients [31,32].

Though there is not yet a global consensus on the utility of circulating cell-free DNA (cfDNA) in cancer detection, our previous study [28], the first of its kind in the sub-region, revealed that breast cancer patients have higher serum cfDNA concentrations than their non-cancer counterparts. The present study further explored the utility of cfDNA in assessing the response to chemotherapy and its potential as a monitoring biomarker, as has been pioneered in rectal cancers [33].

## 2. Materials and Methods

### 2.1. Ethical Approval

Approval for the conduct of the study was given by the Ethical and Protocol Review Committee of the College of Health Sciences, University of Ghana (Reference Number: EPRC/DEC/2019, Protocol Identification Number: CHS-Et/M2-5.3/2019–2020). Participation of this study was voluntary, and participants were granted the right to withdraw from the study without consequences at any time. Before their inclusion in the study, informed consent was obtained from the participants.

### 2.2. Study Site, Participants and Participant Recruitment

The study involved females who had been newly diagnosed with breast cancer as cases and apparently healthy females as controls. Breast cancer patients were recruited from the Department of Surgery of the Korle-Bu Teaching Hospital (KBTH), Accra, Ghana. Controls were recruited from the immediate environment of the hospital. Newly diagnosed breast cancer patients were participants who had been newly diagnosed with breast cancer based on relevant pathology and were treatment-naive. Patients with multiple cancers were excluded from the study. By convenience sampling, consenting, newly diagnosed breast cancer patients attending the Surgical Department of the KBTH between December 2019 and March of 2020 were recruited.

### 2.3. Demographic and Clinocopathological Parameters

Demographic and clinicopathological information were obtained directly from participants using a questionnaire and indirectly from hospital folders of breast cancer patients. Clinicopathological parameters included stage and grade of cancer, tumour size, family history of breast cancer, hypertension and type 2 diabetes mellitus (T2DM) status. Demographic information included age, occupation, age of menarche, menopause, smoking and alcohol intake habits.

### 2.4. Anthropometric Parameters

#### 2.4.1. Waist to Hip Ratio (WHR)

A calibrated tape measure was used to determine hip and waist circumferences to compute (waist circumference (cm)/hip circumference (cm)) as WHR ratio. Waist circumference was determined at the level of the navel to encircle the trunk through the midpoint between the lower rib margin and the iliac crest in a horizontal plane. The hip circumference connected the two greater trochanters laterally, the posterior-most protrusion of the buttocks and anteriorly, midway the mons pubis superior to the pubis symphysis.

#### 2.4.2. Body Mass Index (BMI)

BMI was calculated by using the formula (weight (kg)/height (m)^2^). Participants stood in the anatomical position for their height to the determined using a stadiometer. To determine weight, participants mounted a digital bathroom scale and the readings were recorded.

### 2.5. Blood Samples Collection and Serum Preparation

Venous blood samples (5 mL) were taken at two different time points from the breast cancer patients. Timepoint one (T1) was before the commencement of treatment when the patients had not been exposed to chemotherapy. Timepoint two (T2) was the twelfth week after the commencement of chemotherapy treatment. This period coincides with the end of the third cycle of chemotherapy and is usually when clinicians assess patients’ response to treatment or the efficacy of the chemotherapeutic agent. Venous blood samples were also collected from the apparently healthy volunteers as controls. Samples of venous blood were taken from the median cubital vein of each participant into labelled serum separator tubes. Fifteen (15) minutes thereafter, each sample was centrifuged at 1000× *g* for 15 min to obtain the serum. The serum was removed and aliquoted for storage at −20 °C until use.

### 2.6. Sample Preparation for Quantitative PCR (qPCR)

Frozen serum samples were thawed at room temperature on ice. The sample preparation for qPCR was carried out according to the method described by Iqbal et al. [34]. A preparation buffer containing 2.5% Tween-20, 50 mmol/L Tris, and 1 mmol/L EDTA was constituted. To each 20 μL of serum sample, 20 μL of the preparation buffer was added to deactivate proteins that bound to template DNA or DNA polymerase which could invalidate qPCR results. To the mixture, 20 μg of proteinase K (Inqaba Biotec, Pretoria, South Africa) was subsequently added for protein digestion at 56 °C for 50 min followed by heat inactivation at 95 °C for 5 min. The mixture was centrifuged at 1000× *g* for 5 min and 0.5 μL of the supernatant was used as template for each qPCR reaction.

### 2.7. qPCR Conditions and Quantification of ALU Fragments

For each direct qPCR, the reaction mixture consisted of 0.5 µL of template sequence, 0.5 µL of forward and reverse primers each, for ALU115 and ALU247, 6.25 µL of the master mix (LUNA MASTERMIX) and 4.75 µL of distilled water, together making up a total reaction volume of 12.5 µL. Real-time qPCR amplification was performed on each plate with an initial denaturation at 95 °C for 60 s, followed by 45 cycles of denaturation at 95 °C for 15 s, annealing at 60 °C for 30 s and extension for 60 °C for 30 s using Quantstudio 5^®^ real time PCR. For each plate, a negative control (sample without targetable DNA) was performed. The DNA amplification targeted the consensus ALU sequences (ALU115 and ALU247). Two sets of ALU primers were designed: the primer set for the 115 bp amplicon (ALU115) amplified both shorter and longer DNA fragments, whereas the primer set for the 247 bp amplicon (ALU247) amplified only longer DNA fragments. The sequences of ALU115 primers were forward: 5′-CCTGAGGTCAGGAGTTCGAG-3′ and reverse: 5′-CCCGAGTAGCTGGGATTACA-3′; ALU247 primers were forward: 5′-GTGGCTCACGCCTGTAATC-3′ and reverse: 5′- CAGGCTGGAGTGCAGTGG-3′. β-actin was used as normalizer for all qPCR assays. The sequences of β-actin primers used were forward: 5′-GACCTCTATGCCAACACAGT-3′ and reverse: 5′-AGTACTTGCGCTCAGGAGGA-3′.

### 2.8. DNA Integrity Determination

DNA integrity was calculated as the ratio of ALU247 to ALU115. ALU115 values represented the total amount of free serum DNA. ALU247 values represented the total amount of DNA released from non-apoptotic cells.

### 2.9. Data Analysis

Demographic and clinical parameters were captured and validated using Microsoft Excel. The results of both ALU 115 and 247 were log transformed before undertaking any statistical analysis. This was because the untransformed gene expression values were not normally distributed but skewed.

The results were then analysed statistically using Statistical Package for Social Sciences version 20 statistical software for Windows. All parametric data were expressed as the mean and standard deviation. Statistical significance of the differences between three or more group means were performed by using one-way analysis of variance (ANOVA). A paired Student *t*-test was used to compare the differences between repeated measures (before chemotherapy and after the third cycle of chemotherapy of the breast cancer patients). Differences in the study parameters between groups (breast cancer patients and apparently healthy controls) were assessed using an unpaired Student *t*-test. Spearman’s rank correlation coefficient was used to find association between ALU 115 and ALU 247 concentrations with age, BMI and WHR. A *p*-value ≤ 0.05 was considered as significant.

## 3. Results

### 3.1. Demographic Characteristics of Study Participants

A total of 64 females, 32 breast cancer patients (cases) and 32 apparently healthy individuals (controls) were recruited into the study. The mean ages of the cases and the controls, respectively, were 50.84 ± 12.41 years and 61.69 ± 14.74 years (*p* = 0.0023). The menarche index was statistically higher in the healthy individuals as compared to the breast cancer patients, with mean ± SD of 16.38 ± 1.31 and 12.63 ± 2.20, respectively (*p* < 0.0001). Of the 32 breast cancer patients, five (15.6%) routinely consumed alcohol whereas three (9.4%) of the apparently healthy individuals did (Table 1). For the cancer patients, 21 (65.5%) and 20 (62.5%) were hypertensive and diabetic, respectively whereas the controls reported nine (28.1%) and five (15.6%), respectively. None of the groups recorded any smokers. Compared to the one (3.1%) control who had family history of breast cancer, more than half (56.3%) of the cancer patients had a history of the disease in their families.

### 3.2. BMI and WHR among Groups

From Table 2, the BMI was significantly (*p* = 0.0001) higher in the breast cancer group compared to the control group with mean ± SD of 31.03 ± 7.52 and 24.35 ± 5.19, respectively. However, WHR did not vary among the groups (*p* = 0.375).

### 3.3. Clinicopathological Characteristics of Breast Cancer Cases

The clinicopathological information on breast cancer patients is summarized in Table 3. All the 32 breast cancer cases were invasive ductal carcinoma of no special type. Out of those, 18 (56.3%) were cancer grade III while 12 (37.5%), 1 (3.1%) and 1 (3.1%) were cancer grade II, grade IV and grade I, respectively. Additionally, 4 (12.5%), 25 (78.1%), and 3 (9.4%) of the cases were, respectively, confirmed as stage II, III and IV. Regarding the molecular subtypes of breast cancer, the frequencies among the cases were 6 (18.7%), 10 (31.2%), 7 (21.8%) 9 (28.1%), and for luminal A, luminal B, HER2-enriched and triple negative, respectively.

### 3.4. Serum cfDNA Concentrations and cfDNA Integrity among the Breast Cancer Patients and Apparently Healthy Controls

Before the initiation of chemotherapy (timepoint 1, T1), the levels of ALU 115 and 247 were significantly higher in the breast cancer patients than in the controls (Table 4). Though DNA integrity was higher in the breast cancer patients than the controls, the difference was not significant (*p* = 0.522). After the third cycle of chemotherapy treatment (timepoint 2, T2), the concentrations of ALU 115 and ALU 247 significantly decreased in the breast cancer patients (Table 5). In terms of quantum, DNA integrity at T2 was higher compared with that at T1, though the difference was not significant.

### 3.5. Serum cfDNA Concentrations among Tumour Parameters in the Breast Cancer Patients at Timepoint 1 and Timepoint 2

For tumour parameters, molecular subtype, stage and grade, concentrations of T2 values for ALU 115 and ALU 247 were lower compared to those for T1 (Table 6), though the majority of the time, the differences between the means were not statistically significant. Similarly, DNA integrity when computed was higher at T2 than at T1, although the inter-mean differences were not significant.

## 4. Discussion

This study assessed the utility of serum cfDNA as a blood biomarker for evaluating the response to chemotherapy in breast cancer patients. Conforming with earlier reports [18,25,26,27], serum cfDNA levels were higher in breast cancer patients compared to controls despite the cancer patients being significantly younger than the cases (Table 1). Aging is associated with cellular stress and is accompanied by alterations to the number of apoptotic cells and DNA release, which results in elevated serum cfDNA levels [35,36]. Consequently, older individuals are reported to exhibit higher levels of circulating cfDNA compared with younger individuals [35,37]. Our present report strongly buttresses reports that cancers raise serum cfDNA concentrations. It would have been expected that the older controls would have demonstrated higher serum cfDNA levels than the relatively younger cancer patients. On the contrary, the relatively younger cancer patients demonstrated elevated serum levels of cfDNA compared to the younger controls and thus, erasing the possibility of age being a confounder. The elevated serum cfDNA levels in the cancer patients may be attributed to the presence of the cancer. Among the breast cancer patients, serum cfDNA levels were significantly reduced after the third cycle of treatment (T2) compared to the levels before the commencement of chemotherapy at T1.

Evaluating established risk factors associated with breast cancer development including the onset of menarche, BMI, WHR, smoking and alcohol consumption, our results showed cancer patients recorded earlier onset of menarche (12.63 ± 2.20 years) compared to controls (16.31 ± 1.31 years). The early onset of menarche is a known risk factor for breast cancer [38] and it is suggested to contribute to cancer development due to the prolonged period of exposure to oestrogen. The early onset of menarche makes women have more menstrual cycles, particularly before age 14, and thus increases their lifetime exposure to oestrogen, contributing to a slightly higher risk of breast cancer [39,40].

Exposure to oestrogen is a significant risk factor of breast cancer as oestrogen has been reported together with its catechol metabolites as carcinogens in different tissues, including the kidneys, liver, uterus, and mammary glands [41]. Oestrogen physiologically acts upon other cells to increase their proliferative capacity or cellular hyperplasia, speeding up and increasing the production of any present mutant cancerous cells [39,40,41,42].

BMI and WHR estimate body fat accumulation with higher values associated with higher risk of the development of certain diseases, including cardiovascular diseases and breast cancer [40,43,44]. It is therefore not surprising that a majority of the breast cancer patients were hypertensive (Table 1), which has also been implicated as a risk factor for breast cancer [45].

The mean BMI of the breast cancer patients was 31.03 ± 7.52 kg/m^2^ and that of the controls was 24.35 ± 5.19 kg/m^2^. The majority of the breast cancer patients were obese, while the apparently healthy individuals had a normal BMI. Obesity increases the risk of breast cancer and every 5 kg/m^2^ increase in BMI correlates to a 2% increase in the risk of breast cancer in females [40,44]. Higher BMI and WHR translate into increased amounts of fat deposition in the body. Higher BMI and WHR are breast cancer risk factors partially because fat cells are oestrogen-producing, and extra fat cells mean more oestrogen in the body [46]. In our study, WHR did not significantly vary among the groups and may therefore suggest that in the age group assessed BMI but not WHR associated with breast cancer.

Sixty-two per cent (62.5%) of the breast cancer patients were diabetic as compared to 15.6% of the controls (Table 1). Diabetes causes an increase in blood glucose levels and has been noted as a risk factor for breast cancer [47,48,49].

The serum levels of both ALU 115 (short fragment representing total circulating cfDNA) and ALU 247 (long fragments representing circulating cfDNA released by dying cells, both apoptotic and necrotic) were elevated in breast cancer patients compared to healthy controls (Table 4). This is most likely the result of increased amounts of truncated fragments of DNA released by both apoptotic cells and tissue necrotic activities of the body tissues into the bloodstream of breast cancer patients [28,29]. DNA integrity was higher in breast cancer patients with a mean of 1.22, compared to controls with a mean of 1.07. DNA integrity was lower in healthy individuals, probably due to low necrotic activity in body tissues, thereby lowering the concentration of longer DNA fragments in the bloodstream.

The concentration of ALU 115 and ALU 247 significantly decreased after the third cycle of chemotherapy (T2) with increased cfDNA integrity (Table 5), an observation readily attributable to the effect of chemotherapy and the body’s cfDNA clearance system. Chemotherapy aims at destroying cells (especially fast dividing cancerous cells) inadvertently inducing the release of cellular DNA from necrotic tissues into the bloodstream and consequently elevating the levels of cfDNA in the blood [50]. At time T2 (after the third cycle of chemotherapy), when the efficiency of the chemotherapeutic agent is assessed, it is expected that, if the patient responded positively to the chemotherapeutic agent, the tumour dimensions should have reduced since an appreciable number of the cancerous cells would have been destroyed. With the heavy destruction of cells and the concomitant rise in serum cfDNA levels, it is expected that T2 cfDNA values should be higher than T1 values, but the opposite was rather observed. T2 values lower than T1 values are suggestive of (1) a reduction in the population of cancer cells and/or their activities [51], and (2) a robustly efficient and uncompromised cfDNA clearance system [52]. Regarding the latter, the increased concentration of circulating cfDNA (specifically the longer DNA fragments) as a sequel to the activity of chemotherapy agent may have probably activated the cfDNA clearance system to clear off the released DNA from the bloodstream [52] and thus, effectively decreasing the concentrations of cfDNA in the blood. However, due to the excessive destruction of the cancer cells by the chemotherapy, more necrotic DNA (longer DNA fragments) may have been released into the bloodstream. The elevated levels of the necrotic DNA (longer DNA fragments) consequently might have overloaded the clearance system, causing the accumulation of the longer uneven fragments of DNA in the blood, therefore, affecting the DNA integrity [53]. The observed trend in serum cfDNA levels between T1 and T2 was not discriminatory to tumour grade and stage, and molecular subtypes. In most cases, T2 values reduced either significantly or marginally, while in a few of the cases, the values remained unchanged (Table 6). The limitations of this pilot study include: (1) the small sample size which was influenced by a significant level of attrition on the part of the recruited participants who could not afford the costs related to chemotherapy and thus dropped out of the study, (2) the ability to detect the level of tumour shrinkage after the third cycle of chemotherapy, and (3) failure to collect data on the specifics and details of the drug each patient received to determine the drug with the greatest effect on cfDNA. Future expanded study will involve a larger sample size, the determination and correlation of tumour size after the third cycle of chemotherapy with cfDNA levels, and details of chemotherapy administered to individual participants.

## 5. Conclusions

We report that after the third cycle of chemotherapy, serum cfDNA levels reduced compared to the levels before the commencement of treatment. This may provide a relatively less expensive and very minimal invasive procedure to evaluate the response of breast cancer patients to chemotherapy. Further studies should advance this study and determine justifiable scientific and clinical ranges for cfDNA levels for clinical utility.

## Figures and Tables

**Table 1 medsci-09-00037-t001:** Socio-demographic characteristics of study population.

**Variable**	B**reast Cancer Patients** **(*n* = 32)**	**Controls** **(*n* = 32)**	**95% CI of Difference**	***p*-Value**
Age (yrs)	50.84 ± 12.41	61.69 ± 14.74	−17.66–(−4.04)	0.0023 *
Menarche (yrs)	12.63 ± 2.20	16.38 ± 1.31	−4.65–(−2.85)	<0.0001 *
Menopause (yrs)	47.45 ± 4.27	47.33 ± 4.55	−2.01–2.33	0.4172
**Variable**	**Breast Cancer Patients (*n* = 32)**	**Controls** **(*n* = 32)**	**Chi-Square (χ^2^)**	***p*-Value**
*Alcohol intake*				
Yes	5 (15.6%)	3 (9.4%)	0.5714	0.4497
No	27 (84.4%)	29 (90.6%)		
*Smoking*				
Yes	0 (0.0%)	0(0.0)		-
No	32 (100%)	32 (100.0%)		
*Hypertensive*				
Yes	21 (65.6%)	9 (28.1%)	7.592	0.005 *
No	11 (34.4%)	23 (71.9%)		
*T2DM*				
Yes	20 (62.5%)	5 (15.6%)	12.866	0.0003 *
No	12 (37.5%)	27 (84.4%)		
*Family History BC*				
Yes	18 (56.3%)	1 (3.1%)	19.163	<0.00001 *
No	14 (43.7%)	31 (96.9%)		

*n* = study population, data are presented as mean ± standard deviation. Chi-square (χ^2^). * *p*-value is statistically significant, T2DM = type 2 diabetes mellitus, BC = breast cancer, CI = confidence interval.

**Table 2 medsci-09-00037-t002:** Anthropometric data of study population.

Variable	Breast Cancer Patients (*n* = 32)	Controls (*n* = 32)	95% CI of Difference	*p*-Value
BMI (kg/m^2^)	31.03 ± 7.52	24.35 ± 5.19	3.45–9.90	0.0001 *
WHR	0.86 ± 0.06	0.85 ± 0.02	−0.01–0.03	0.3746

*n* = study population, data are presented as mean ± standard deviation. BMI = body mass index, WHR = waist-to-hip ratio. CI = confidence interval. * *p*-value is statistically significant.

**Table 3 medsci-09-00037-t003:** Clinicopathological characteristics of breast cancer cases.

Parameters	Frequency (%) *n* = 32
**Tumour stages**	
II	4 (12.5)
III	25 (78.1)
IV	3 (9.4)
**Tumour grades**	
G1	1 (3.1)
G2	12 (37.5)
G3	18 (56.3)
G4	1 (3.1)
**Molecular subtype**	
Luminal A	6 (18.8)
Luminal B	10 (31.3)
HER-2-enriched	7 (21.8)
Triple negative	9 (28.1)
**Histopathological classification**	
Invasive carcinoma	29 (90.6)
Unknown	3 (9.4)
**Location of cancer**	
Left breast	18 (56.2)
Right breast	14 (43.8)

*n* = breast cancer patients, HER-2-enriched = human epidermal growth factor receptor-2.

**Table 4 medsci-09-00037-t004:** Serum cfDNA concentrations and DNA integrity before chemotherapy (T1).

Parameter	Breast Cancer (*n* = 32)	Controls (*n* = 32)	95% CI of Mean	*p*-Value
ALU 115 (ng/mL)	2.24 ± 0.80	1.83 ± 0.65	−0.77–(−0.05)	0.028 *
ALU 247 (ng/mL)	2.73 ± 0.11	1.96 ± 0.85	−1.07–(−0.47)	<0.0001 *
cfDNA integrity	1.22 ± 0.14	1.07 ± 1.31	−0.62–0.31	0.522

Circulating cell-free DNA (cfDNA) levels of breast cancer patients at baseline (before chemotherapy) were compared with apparently healthy control group. * *p*-value ≤ 0.05 is statistically significant. *n* = number of participants, CI = confidence interval.

**Table 5 medsci-09-00037-t005:** Circulating cell-free DNA concentrations and DNA integrity.

Parameter (ng/mL)	Breast Cancer Patients		95% CI of Mean	*p*-Value
	TI	T2		
ALU 115	2.24 ± 0.80	1.67 ± 0.66	−0.94–(−0.21)	0.003 *
ALU 247	2.73 ± 0.11	2.12 ± 0.69	−086–(−0.36)	<0.0001 *
cfDNA integrity	1.22 ± 0.14	1.27 ± 1.04	−0.32–0.42	0.788

T1: before commencement of chemotherapy: T2: after the 3rd cycle of chemotherapy. * *p*-value ≤ 0.05 is statistically significant. CI = confidence interval.

**Table 6 medsci-09-00037-t006:** cfDNA concentrations among tumour parameters in the breast cancer patients at time points 1 (T1) and 2 (T2).

Parameter	*n*	ALU 115 (ng/mL)		*p*-Value	ALU 247 (ng/mL)		*p*-Value
		T1	T2		T1	T2	
*Molecular Subtype*							
Luminal A	6	2.16 ± 0.53	1.70 ± 0.15	0.068	2.26 ± 0.78	2.19 ± 0.37	0.846
Luminal B	10	1.64 ± 0.61	1.61 ± 0.52	0.907	2.00 ± 0.79	1.94 ± 1.08	0.888
HER2-enriched	7	1.80 ± 0.88	1.60 ± 1.12	0.717	2.21 ± 0.27	1.62 ± 1.20	0.228
Triple negative	9	1.84 ±0.28	1.77 ± 0.49	0.715	2.19 ± 0.55	1.96 ± 0.66	0.433
*Histological Subtype*							
Invasive ductal carcinoma	29	1.84 ± 0.58	1.69 ± 0.66	0.331	2.13 ± 0.71	2.01 ± 0.72	0.525
Unknown	3	1.77 ± 0.34	1.38 ± 1.84	0.736	2.01 ± 0.47	1.52 ± 0.79	0.408
*Tumour Stage*							
Stage II	4	1.69 ± 0.82	1.64 ± 0.69	0.929	2.22 ± 0.53	1.81 ± 0.69	0.361
Stage III	25	2.05 ± 0.37	1.67 ± 0.46	0.002 *	2.23 ± 0.88	2.03 ± 0.70	0.378
Stage IV	3	1.77 ± 0.34	1.52 ± 0.79	0.641	2.01 ± 0.47	1.34 ± 1.85	0.576
*Tumour Grade*							
Grade 1	1	1.88	1.80	-	2.05	1.93	-
Grade 2	12	1.74 ± 0.68	1.56 ± 0.88	0.581	2.24 ± 0.29	1.86 ± 0.94	0.195
Grade 3	18	2.41 ± 0.48	1.83 ± 0.43	0.005 *	2.97 ± 0.87	2.48 ± 0.81	0.089
Grade 4	1	1.39	0.60	-	2.48	1.50	-

Results were presented as mean ± SD: * *p*-value ≤ 0.05 is considered significant: T1 = before commencement of chemotherapy: T2 = after the 3rd cycle of chemotherapy.

## Data Availability

All data generated and analysed during this study are included in this published article.

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
