# Peer review of "Reduced Serum Circulation of Cell-Free DNA Following Chemotherapy in Breast Cancer Patients"

_medsci, 2021, doi:10.3390/medsci9020037_

Round 1
Reviewer 1 Report
In this manuscript, the authors tried to established the levels of serum cfDNA as a proxy for evaluating the response to chemotherapy in breast cancer patients. They reported that after the third cycle of chemotherapy, serum cfDNA levels were reduced compared to the levels before the commencement of treatment, making Serum cfDNA a relatively inexpensive and minimally invasive procedure to evaluate chemotherapy response. The study is significant and sets this correlation very well, and will prove helpful for the clinicians. As the weak point, as the authors themselves acknowledged, the sample size is small, but future studies using this as the pilot studies may involve more patients to reinforce these findings further. Nevertheless, the statistical significance of this study is robust, making it an important one. I want the authors to address these points.
- Please compare and contrast a bit more in detail other clinical procedures in use and publications that evaluate chemotherapy response in breast cancer compared with this study.
- Did the authors follow up on the patients and determine if the tumor's shrinkage after the third cycle of the chemotherapy is correlated with cfDNA levels? Any data on tumor size at the time of diagnosis and after the third cycle of chemo available for these patients from this study?
- Which are the other cancers apart from breast cancer where cfDNA levels predict chemotherapy's outcome based on the literature and clinical practices (if any)?
- The period is missing in line 23 after concentrations.
- Line 47- Is it late detection or early detection?
- Apart from the discussion section, a couple of lines about the significance of ALU 115 and 247 levels should be added in the introduction.
- Please mention more in detail the future directions for this study.
Author Response
- Please compare and contrast a bit more in detail other clinical procedures in use and publications that evaluate chemotherapy response in breast cancer compared with this study.
Response:
Authors have clearly stated how response to chemotherapy are assessed clinically and the possible challenges. Also, a reference has been provided on the size of tumour as clinical assessment procedure.
“The assessment of the effectiveness of chemotherapy is commonly done following the third cycle of treatment (usually about 9 weeks) from the commencement and sometimes may require repeated biopsies for histopathological confirmation of tumour response to treatment [16,17]. This procedure comes with inherent challenges as delays (unintentional) in the administration of suboptimal drugs may lead to disease progression and worsened prognosis. Another assessment is the size of the tumour after chemotherapy, and the challenge is that the size may not accurately reflect the tumour response leading to incorrect clinical decisions [18].
- Did the authors follow up on the patients and determine if the tumor's shrinkage after the third cycle of the chemotherapy is correlated with cfDNA levels? Any data on tumor size at the time of diagnosis and after the third cycle of chemo available for these patients from this study?
Response:
The cross-sectional study did not follow up the patients for tumour shrinkage, therefore no data to establish correlation between tumour size and cfDNA. This preliminary study was to confirm the changes of cfDNA levels in Ghanaian breast cancer patients. The confirmation will warrant large scale study to promote its application in breast cancer monitoring. It is acknowledged as one of the limitations of the study.
- Which are the other cancers apart from breast cancer where cfDNA levels predict chemotherapy's outcome based on the literature and clinical practices (if any)?
Response:
Although there is an accumulated evidence of changes in cfDNA levels in patients and as a potential biomarker for cancer prognosis, it has not been put into clinical practice to monitor chemotherapy. The closest is a study in rectal cancers reported by Agostini et al 2011. This has been included in the manuscript (Reference 31)
- The period is missing in line 23 after concentrations.
Response:
The period has been fixed in the manuscript.
- Line 47- Is it late detection or early detection?
Response: It is late detection. The sentence is correct.
- Apart from the discussion section, a couple of lines about the significance of ALU 115 and 247 levels should be added in the introduction.
Response: This has been done and indicated in the manuscript
- Please mention more in detail the future directions for this study.
Response: This has been done and indicated in the manuscript
Reviewer 2 Report
My first key concern is the lack of data support for the statement in the title that cfDNA reduction “predicts to chemotherapy”.
The ALU247-to-ALU115 ratio should be smaller than 1.0. However, in Tables 4, 5, and 6, the readings for A247 are higher than A115. This draws into question the validity of the key data presented in this manuscript.
The data presented in Table 6, under “histological subtype” that covers all 32 cases of invasive ductal carcinoma, the ALU values and p-values should essentially be identical to those in Table 5, which covers all 32 cases. The discrepancy is puzzling.
Author Response
My first key concern is the lack of data support for the statement in the title that cfDNA reduction “predicts to chemotherapy”.
Response:
The title has been modified to “Serum Circulation Cell-Free DNA Levels Reduce after 3rd Cycle of Chemotherapy in Breast Cancer Patients”
The ALU247-to-ALU115 ratio should be smaller than 1.0. However, in Tables 4, 5, and 6, the readings for A247 are higher than A115. This draws into question the validity of the key data presented in this manuscript.
Response:
The argument is one of the positions of researchers in the field. Proponents argue that DNA integrity is "1" if template DNA is not truncated and "0" if DNA is completely truncated to fragments smaller than 247 bp (base pair) since the annealing sites of ALU115 are within the ALU247 annealing sites. But as the field expands, there have been several reports suggesting that DNA integrity could be greater than 1. Our report is similar to several other observations exampled by the following publications.
- Sriram KB, Relan V, Clarke BE, Duhig EE, Windsor MN, Matar KS, Naidoo R, Passmore L, McCaul E, Courtney D, Yang IA, Bowman R V., Fong KM (2012) Pleural fluid cell-free DNA integrity index to identify cytologically negative malignant pleural effusions including mesotheliomas. BMC Cancer 12:428 . https://doi.org/10.1186/1471-2407-12-428
- Agostini M, Pucciarelli S, Enzo MV, Del Bianco P, Briarava M, Bedin C, Maretto I, Friso ML, Lonardi S, Mescoli C, Toppan P, Urso E, Nitti D (2011) Circulating cell-free DNA: A promising marker of pathologic tumor response in rectal cancer patients receiving preoperative chemoradiotherapy. Ann Surg Oncol 18:2461–2468 . https://doi.org/10.1245/s10434-011-1638-y
- Tuchalska-CzuroÅ„ J, Lenart J, Augustyniak J, Durlik M (2020) Clinical value of tissue DNA integrity index in pancreatic cancer. Surgeon 18:269–279 . https://doi.org/10.1016/j.surge.2019.10.008
The data presented in Table 6, under “histological subtype” that covers all 32 cases of invasive ductal carcinoma, the ALU values and p-values should essentially be identical to those in Table 5, which covers all 32 cases. The discrepancy is puzzling.
Response:
This has been rectified. There were 3 cases whose histological classification could not be ascertained and thus reported as “unknown”. However, in the reporting, that group was mistakenly omitted. This has been corrected make the Invasive Ductal carcinoma group 29 and the Unknown group 3.